# Rebuilding Social Capital through Osekkai Conferences in Rural Communities: A Social Network Analysis

**DOI:** 10.3390/ijerph19137912

**Published:** 2022-06-28

**Authors:** Ryuichi Ohta, Koichi Maiguma, Akiko Yata, Chiaki Sano

**Affiliations:** 1Community Care, Unnan City Hospital, Daito-Cho, Unnan 699-1221, Japan; 2Department of Law and Economics, Faculty of Law and Literature, Shimane University, 1060 Nishikawatsu Cho, Matsue 690-8504, Japan; zzkuma@soc.shimane-u.ac.jp; 3Community Nurse Company, 422 Satokata, Kisuki-Cho, Unnan 699-1311, Japan; yataakiko0425@gmail.com; 4Department of Community Medicine Management, Faculty of Medicine, Shimane University, 89-1 Enya Cho, Izumo 693-8501, Japan; sanochi@med.shimane-u.ac.jp

**Keywords:** Osekkai, social prescribing, social capital, community activity, social network analysis, isolation

## Abstract

Social prescribing can promote the creation of new relationships, which may then promote the building of social capital in communities. One example of a social prescribing tool in Japan is Osekkai conferences, which increase social participation and mitigate the degree of loneliness in rural communities. A clarification of the changes in social interaction and intensity of connections among people through Osekkai conferences could contribute to better social prescribing in rural communities. This social network study was conducted among people who have participated in an Osekkai conference. The primary outcomes of degrees and centrality were measured as the degree of social capital. The primary outcomes were compared between April and September 2021 and between October 2021 and March 2022. The continuous performance of Osekkai conferences as social prescribing tools led to an increase in conference participation, mainly by middle-aged women in the communities. Based on a social network analysis, the average direct connection with each person did not increase; the network density decreased gradually; the network diameter decreased from 6 to 5. Regarding the node-level statistics, harmonic closeness centrality and eccentricity decreased, and modularity increased. Social prescribing initiatives should focus on improving social capital in communities, which may improve the number and meaningfulness of the collaborations among organizations and indigenous communities.

## 1. Introduction

Within community settings, social prescribing can promote the creation of new relationships, potentially building up social capital. Social capital refers to the networks of relationships among people who live and work in a particular society, enabling that society to function effectively and leading to better conditions for individuals and communities [1,2]. Social capital is essential for the sustainability of communities, and high levels of social capital could be related to better health and quality of life [3,4]. Social capital should be enhanced to ensure sustainability and management of health conditions such as chronic diseases [5,6,7]. The aging societies has increased health care problems among older populations causing the problems of multimorbidity and polypharmacy [8,9,10]. These issues should be solved by involving not only health care professionals but also local people [11].

Social prescribing refers to a means by which healthcare professionals seek to address the nonmedical causes of ill health with nonmedical interventions [12,13]. Through social prescribing, people can connect with various resources (e.g., healthcare providers and institutions), possibly providing more opportunities for interactions and collaborations among people [12]. Within the context of communities, interactions and collaborations among people facilitate the development of new relationships, potentially contributing to productive relations, such as those associated with health management. For example, a study shows that the established relationship among people in communities provided people suffering with specific health problems with supportive information regarding their physical and mental issues [12,13]. This supportive information, in turn, enabled people suffering with such issues to perform self-management and receive social support and medical care effectively, which could enhance their quality of life [14,15]. This relationship can also be referred to as “social capital.” Another study shows that social prescribing can improve the social capital and health conditions of communities [12].

Social prescribing is performed worldwide, although the strategies vary by the context and culture of countries. There are various indigenous activities related to helping other people that are rooted in various Asian cultures [12,13]. One social prescribing intervention particular to Japan is the so-called Osekkai conference, which can increase social participation and mitigate the degree of loneliness in rural communities [16,17]. In such conferences, participants deal with social and healthcare issues in their communities and connect community resources to each of these issues [16]. Further, various people participate and discuss matters, which could promote valuable connections among them and the development of new social capital. These conferences are held in rural Japanese communities and have spread all over Japan [18]. In sum, they aim at promoting relationships and enriching the social capital in rural communities through the creation of collaboration opportunities [16,17,18].

To the best of our knowledge, no study thus far has analyzed the changes in social interactions and the intensity of the connection among people in communities promoted through Osekkai conferences. Thus, we posit the following research question: “Do Osekkai conferences, as social prescribing tools, increase social connections in communities?” The increase in connections promoted by Osekkai conferences may facilitate the creation of new networks and opportunities to establish social capital [19,20]. In summary, this study aimed to investigate changes in the connections among the participants of an Osekkai conference through a social network analysis (SNA). By clarifying these changes and its effects on the structure of these relationships, this study helps invested stakeholders in advancing the use of Osekkai conferences as social prescribing tools in rural communities.

## 2. Materials and Methods

This study used SNA to assess the changes in the degree of community interaction through participation in Osekkai conferences [21]. This study went on from April 2021 to March 2022.

Unnan City, in the Shimane Prefecture, is a rural Japanese area. In 2020, the total population of Unnan was 37,638 (18,145 men and 19,492 women), with 39% being over 65 years old; this proportion is expected to reach 50% by 2050 [22]. There are 30 autonomous community organizations in Unnan City, each of which has various functions for managing their respective social issues (e.g., social isolation, accessibility to medical care, and succession of traditional activities). Each district has at least one autonomous community organization [22].

Various rural individuals participate in Osekkai conferences, and some of the participants have different professional backgrounds (e.g., medicine, law, public health, rehabilitation, architecture, community development, and transportation). Partakers work collaboratively to solve their community’s problems by sharing their own experiences and suggesting solutions, generating an Osekkai conference plan. This plan is then carried out in their community. The results of the plan are shared in the following Osekkai conference. Through continuous discussions, Osekkai conference plans can be revised and the quality of care in the community can be improved [18]. Subsequently, the discussions can generate new, effective collaborations between participants with different resources. Through these collaborations, new relationships can be established, and the continual implementation of Osekkai conferences may improve social participation and diminish loneliness in the community [18]. The framework of Osekkai conferences is represented in Figure 1 [11,16,17].

The participants of this study were people who were interested and participated in Osekkai conferences or who facilitated and performed Osekkai conferences. The information regarding an Osekkai conference is provided at the Unnan City Hall via social media and the city’s local newspapers. People interested in the conference send a mail or make a call to the office responsible for the conference to register as participants. Regarding participants’ activities related to an Osekkai conference, two roles were focused on in this study: those implementing an Osekkai conference and those participating in an Osekkai conference. The citizens who hoped to participate in these conferences went to the community center, where the conference was held on a previously established day.

This study used an SNA to analyze the change in social connections among participants of an Osekkai conference. An SNA is a structured process for investigating social structures using concepts established from network analyses and graph theory [21]. Using this method, researchers can visualize and analyze social connections by measuring the strength of relationships between members of the network and the network’s density.

The participants’ names, gender, age, place of residence, and their role in the conference and provision of Osekkai conferences were measured at the beginning of each conference and the provision of Osekkai. The SNA dataset was created from the Osekkai conference participants’ list. The network was described using the level of each Osekkai conference and participants by using Gephi software. In graphing, the participants who organized the Osekkai conference and provision of Osekkai were considered a source of connection, and participants of the Osekkai conference and provision of Osekkai as a target of the connection. Each participant was coded anonymously before the analysis.

We reported the descriptive statistics of the number of participants in the Osekkai conference, of Osekkai, and participants’ demographics. Second, the SNA was used for a relational matrix representing the network. These are presented as a sociogram, with the periods of the conferences as a source of connection and the links between the participants in the network. We used the SNA software Gephi to generate sociograms and network-wide node (participant-levels) statistics/metrics and visually depict the network structure. To demonstrate how the connections among participants evolved during the study period, we explored the connections of all participants (nodes) at two times, developing a visual representation for the end of September 2021 (Figure 2) and the end of March 2022 (Figure 3). To focus on the connections among the participants, the following statistics were calculated in a nondirectional way in the SNA. From a numerical perspective, a set of metrics was calculated to describe the status of the entire network, including metrics calculated at the node level to describe the characteristics of each participant represented in the network. Statistical significance was defined as a *p*-value < 0.05.

### 2.1. Specific Metrics of Interest at the Network Level

The average degree describes how connected a typical participant is within the network. It is calculated by taking the average number of connections for each participant. Regarding the direction of the connection, the degree is differentiated into indegree and outdegree. The network density describes the number of connections between participants. In the context of this study, a denser network would mean participants are more directly connected to each other, while a less dense network would mean fewer connections. Network density is calculated as the proportion of actual connections to all possible connections (range: 0–1). The network diameter is the length of the longest of all the computed shortest paths between all pairs of nodes in the network.

### 2.2. The Specific Metrics of Interest at the Node Level

Eigenvector centrality measures a node’s importance while considering the importance of its neighbors. It assigns relative scores to all nodes in the network based on the concept that connections to high-scoring nodes contribute more to the score of the node in question than equal connections to low-scoring nodes.

Closeness centrality measures how close a node is to all the other nodes in a network and whether the node is located on the shortest path between other nodes. A high closeness centrality means that there is a large average distance to other nodes in the network.

Harmonic closeness centrality is a variant of closeness centrality that was developed to solve the problem the original formula had when dealing with unconnected graphs. The closeness centrality is calculated, including unconnected graphs.

Betweenness centrality is measured based on the number of shortest paths between any two nodes passing through a particular node. Nodes around the edge of the network would typically have a low betweenness centrality, and a high betweenness centrality might suggest that the individual is connecting various parts of the network together.

Measuring eccentricity captures the distance between a node and the furthest node from it. A high eccentricity means that the furthest node in the network is a long way away, and a low eccentricity means that the furthest node is quite close.

Measuring modularity captures the structure of networks or graphs based on each node connection. It was designed to measure the strength of the division of a network into modules (also called groups, clusters, or communities). Networks with high modularity have dense connections between the nodes within modules but sparse connections between nodes in different modules.

### 2.3. Ethical Considerations

The participants’ anonymity and confidentiality were ensured throughout the study. All participants provided written informed consent before participating in the conferences and answering the questionnaire. All procedures in this study were performed in compliance with the Declaration of Helsinki and its subsequent amendments. The Unnan City Hospital Clinical Ethics Committee approved the study protocol (no. 20220021).

## 3. Results

### 3.1. The Demographic Data

Throughout the duration of this study, the number of participants increased from 567 to 1098. The rate of female participants increased from 55.9% to 67.9%. Most participants were between 30–40 years old and from Unnan City. Between the periods, the rate of female participants increased by a statistically significant extent. There were no significant differences in age, performing Osekkai, or living places (Table 1).

### 3.2. Change in the Network over Time

In the network-level statistic, average degree, indegree, and outdegree did not statistically change over the study period. The network density decreased gradually, and the network diameter decreased from 6–5. Regarding the node-level statistic, Eigenvector centrality, closeness centrality, and between centrality did not change statistically. In contrast, harmonic closeness centrality and eccentricity decreased, and modularity increased statistically (Table 2).

## 4. Discussion

The findings from the SNA demonstrated that continuously performing Osekkai conferences (as social prescribing tools) and Osekkai plans was associated with an increase in the number of conference participants, mainly young to middle-aged women, in the communities. Based on the SNA, each person’s average number of direct connections did not increase, but the network density decreased gradually, and the network diameter decreased from 6 to 5. This demonstrates an increase in indirect connections among the participants. Regarding the node-level statistics, harmonic closeness centrality and eccentricity decreased, and modularity increased, indicating that each participant could increase their social capital, regarding the strength of their connection with others, in their communities.

This study shows that more women participated in the conferences throughout the duration of the study, mainly those aged between 20–40 years. Hence, social prescribing tools such as Osekkai conferences could drive the participation of women based on their demographic characteristics. This trend could be affected by rural female characteristics [22]. Two prior studies show that rural women have a greater tendency to converse regarding their lives and share their daily life difficulties and potential solutions compared with men [18,22]. Further, three studies show that Osekkai conferences might give women in rural communities the opportunities for engaging in such conversations, something that may have been lost amid and due to the COVID-19 pandemic [18,23,24]. Then, by considering that many of the middle-aged women who partake in these conferences have connections with men in their rural communities, it may be that the dialogues promoted through Osekkai conferences could lead to solutions for not only female but also male social issues gradually and effectively.

By participating in an Osekkai conference, the participants can become connected with others intentionally and find it easier to connect to others in their communities. However, the degree of direct connections that each participant in our study did not change, so the participants might not notice that they acquired various new relationships through an Osekkai conference. In contrast, the network density and diameter both decreased, meaning that the participants could connect to others in the community more easily and quickly. This phenomenon occurred while the number of participants gradually increased. Previous studies have demonstrated that network density can exhibit the community’s strength and nurturing empowerment, which in turn can drive effective community activities [25,26]. Furthermore, indirect connections are important to sustain communities [27,28]. Difficult situations involving individuals can be dealt with by making use of various support functions, which social support programs can provide by promoting indirect connections among people [29,30]. This study demonstrates that social prescribing could improve indirect connections in communities and build new social capital for individuals and communities. Consequently, the strength of the connection with others in the community, as an aspect of social capital, might be nurtured through the continuous implementation of Osekkai conferences.

This study also demonstrates that harmonic closeness centrality and eccentricity decreased, and modularity increased, indicating that each participant could increase their social capital via the strength of the connection with others in their communities. Direct and indirect connections can affect individuals’ lives in communities [27,31], as people can receive help from these two types of connections to lead their lives effectively [32,33]. For example, when an individual needs some resources, such as healthcare or better home conditions, their connections with others can determine their relationship with the needed resources, although they do not directly connect with those resources [18]. Eventually, people with various indirect relationships can reach the resources which they need, which could be described as the social capital used to enrich their lives [34]. Therefore, by participating in an Osekkai conference, as a social prescribing tool, the participants can increase their direct and indirect relationships with others, enriching the social capital in their community.

Based on our results, we believe that the application of Osekkai conferences as social prescribing tools could be broadly expanded by three measures: ensuring their continuity; finding ways to collaborate with other community activities and resources; encouraging the transmission of good health practices using the existing positive connections within the community. First, for the continuity of social prescribing, conference organizers should collaborate with various private and public organizations to acquire funding, information, and resources to perform the conferences [35,36]. In this study, the SNA focused on individual interactions, and obtaining help from local governments and public systems such as the post office should be encouraged [11,37,38]. Future studies could investigate the changes in the collaboration arising from social prescribing and private and public organizations.

Second, other community resources should be used to provide social prescribing. In communities, indigenous community activities that connect people [31,39] can be organized in collaboration with social prescribing [40]; such collaboration can then nurture the relationships in communities more effectively and enable for the rebuilding of social capital [22,41]. Considering that this study was focused on the relationship of the participants of an Osekkai conference, future studies could expand our examinations by focusing on the combination of indigenous community activities and social prescribing. This will enable for a clearer identification of strategies for nurturing relationships and social capital in communities.

Finally, the transmission of good health practices should be encouraged based on the existing positive connections within the community. Indeed, research shows that social connections can transmit not only information but also health behaviors. For example, the obesity and nutrition conditions of an individual could affect their surroundings, then which could increase obesity and the number of unhealthy community members [42,43,44]. Furthermore, older people in rural contexts struggle to deal with their health conditions due to a lack of knowledge and resources [45]. To improve health conditions and behaviors in communities, good health practices should be spread through community relationships. For example, a community-wide exercise campaign could improve healthy behaviors in communities [42,43], and to expand exercise and nutrition, social connections should be used for appropriate information and education [42,43,46]. This study showed that Osekkai conferences can increase the social capital and connections of a community, through which good health practices can then be spread. Finally, these processes could lead to gradual improvements in health practices within the community.

This study has several limitations. It focused on rural Japanese communities, so the study’s setting may represent only rural communities with a lack of medical resources, an aging society, and high numbers of isolated older people. Future studies should focus on a broader range of communities and cultures. Another limitation pertains to the sampling methods. This study focused on participants’ connections within the conferences as judged by whether they participated in a specific conference; that is, we did not assess their specific collaboration to the conference and provision of Osekkai conferences using variables and methods that allowed for the direct assessment of these variables. Future studies should investigate the relationships and interactions in Osekkai conferences that affect the participants’ behaviors related to Osekkai. Furthermore, the study was a single-arm interventional study with no comparison group, affecting internal validity, and the Osekkai conferences (the social prescribing tools) were provided based on previously published studies [16,17,18]. These characteristics of our research highlight a gap for future scholars: to investigate the effect of social prescribing using cluster randomized control trials and to further demonstrate the scientific significance of these conferences.

## 5. Conclusions

The continuous performance of Osekkai conferences in rural communities may serve as social prescribing tools, and this continuity may lead to an increase in the number of conference participants. Based on an SNA, the provision of Osekkai conferences was related to a gradual decrease in network density and diameter. The participants could then increase their social capital using the strength of the connection with others in their communities. Accordingly, social prescribing interventions should focus on improving social capital, which may improve the number and meaningfulness of the collaborations among organizations and indigenous communities as well as help ensure enhanced connections and social capital in the community.

## Figures and Tables

**Figure 1 ijerph-19-07912-f001:**
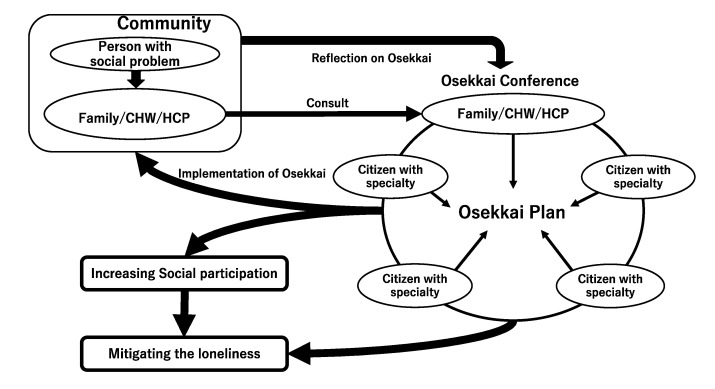
The framework of an Osekkai conference. Notes: CHW, community health worker; HCP, healthcare practitioner.

**Figure 2 ijerph-19-07912-f002:**
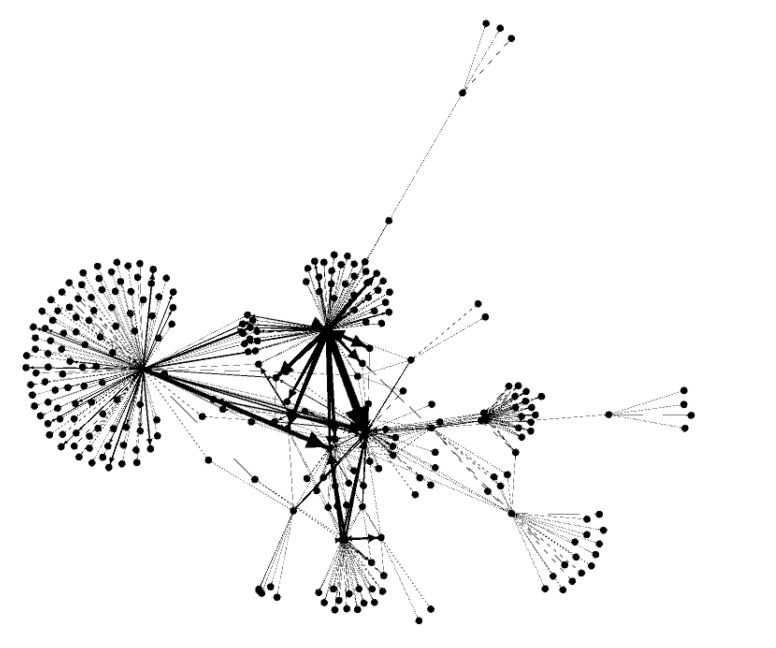
Visual representation of the connection of the participants by September 2021.

**Figure 3 ijerph-19-07912-f003:**
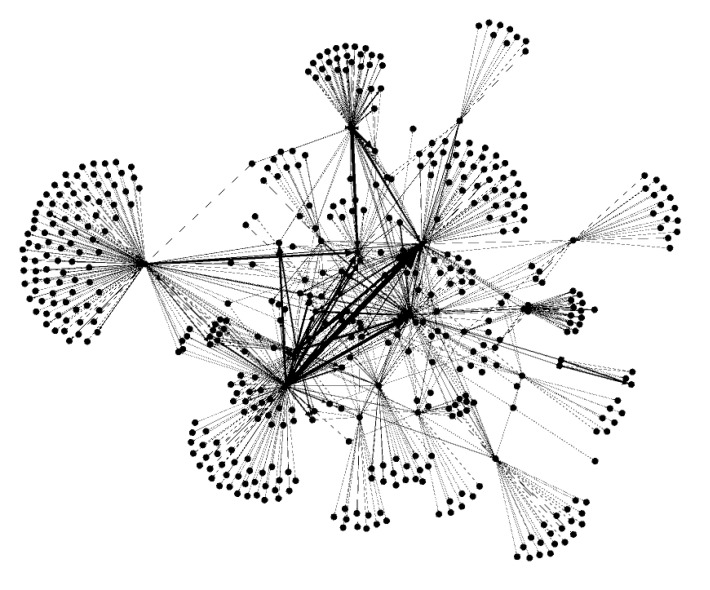
Visual representation of the connection of the participants by March 2022.

**Table 1 ijerph-19-07912-t001:** Participants’ demographic data.

	Timing		
Variables	April to September 2021	April 2021 to March 2022	*p*-Value
Total number of the participants	567	1098	
Male gender (%)	256 (45.1)	352 (32.1)	<0.001
Age (%)			
Under 20	19 (3.4)	50 (4.6)	0.299
20 s	36 (6.3)	56 (5.1)	0.309
30 s	156 (27.5)	271 (24.7)	0.214
40 s	76 (13.4)	175 (15.9)	0.193
50 s	19 (3.4)	25 (2.3)	0.2
60 s	20 (3.5)	49 (4.5)	0.437
70 s	38 (6.7)	72 (6.6)	0.917
80 s	1 (0.2)	2 (0.2)	1
90 s	2 (0.4)	5 (0.5)	1
Unknown	200 (35.3)	393 (35.8)	0.871
Performing Osekkai	57 (10.1)	100 (9.1)	0.537
Living place			
In the city	395 (69.7)	760 (69.2)	0.866
Outside of the city	172 (30.3)	338 (30.8)	

**Table 2 ijerph-19-07912-t002:** The change in variables of the network at the network and node levels.

	Timing		
Factor	April to September 2021	April 2021 to March 2022	*p*-value
Network level			
Average degree (SD)	3.89 (10.45)	3.82 (11.68)	0.931
Indegree (SD)	1.95 (2.14)	1.91 (2.32)	0.829
Outdegree (SD)	1.95 (9.96)	1.91 (10.53)	0.962
Network density	0.014	0.013	
Network diameter	6	5	
Node level			
Eigenvector centrality (SD)	0.08 (0.19)	0.08 (0.10)	0.371
Closeness centrality (SD)	0.33 (0.07)	0.33 (0.06)	0.489
Harmonic closeness centrality (SD)	0.36 (0.08)	0.35 (0.06)	0.007
Between centrality (SD)	287.33 (1641.23)	517.66 (3413.11)	0.292
Eccentricity (SD)	5.35 (0.92)	4.41 (0.59)	<0.001
Modularity (SD)	2.07 (1.35)	3.38 (2.59)	<0.001

## Data Availability

The datasets used and/or analyzed during the current study may be obtained from the corresponding author upon reasonable request.

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
