# Peer review of "Rebuilding Social Capital through Osekkai Conferences in Rural Communities: A Social Network Analysis"

_ijerph, 2022, doi:10.3390/ijerph19137912_

Round 1

Reviewer 1 Report

Thank you for giving me an opportunity to review this paper regarding a social network analysis of rebuilding social capital through Osekkai Conferences in a Japanese context. The article lays out its argument clearly, while the contributions are mainly on the descriptive level. There are some problems that the authors should carefully address:

1. The part of literature review is inadequate and fails to find out the shortcomings of existing research. Although the authors cited some references, this paper lacked a thorough review of studies related to the meanings and possible outcomes of social capital, let alone providing critical comments on this issue.

2. There is no detailed discussion about the relationship between social prescribing and social capital. As authors mentioned, “social prescribing refers to a means by which healthcare professionals seek to address the non-medical causes of ill health with non-medical interventions”. However, the social network that social capital emphasizes does not necessarily correspond to medical services. Thus, the connotations of social prescribing and social capital need to be elaborated more clearly.

3. In the statistical analysis, the authors did not pay enough attention to the heterogeneity of network members. Is there significant difference between male and female groups, or various age groups? If the answer is yes, why gender/age difference occurs? The robustness test of the research results (e.g., harmonic closeness centrality and eccentricity decreased, and modularity increased, the network density in rural communities decreased gradually, and the diameter became smaller) is suggested to be added in order to improve the persuasiveness of the statistical evidence.

In summary, this paper is well organized and provides some original findings. However, there are questions regarding the literature review, clarification of core concepts and analytical strategy.

Author Response

Responses to the reviewers’ comments

Thank you very much for reviewing our manuscript and providing suggestions for its improvement. We have provided point-by-point responses to the reviewers’ comments; our revisions are indicated in red font here and in the document. We hope that the revised manuscript meets the journal’s requirements and can now be considered for publication.

Thank you for giving me an opportunity to review this paper regarding a social network analysis of rebuilding social capital through Osekkai Conferences in a Japanese context. The article lays out its argument clearly, while the contributions are mainly on the descriptive level. There are some problems that the authors should carefully address:

1.The part of literature review is inadequate and fails to find out the shortcomings of existing research. Although the authors cited some references, this paper lacked a thorough review of studies related to the meanings and possible outcomes of social capital, let alone providing critical comments on this issue.

Response:

Thank you for the constructive suggestions. We agree with the suggestions and have added evidence showing the theoretical and conceptual frameworks of social prescribing and social capital in the introduction (lines 44 to 73).

  1. There is no detailed discussion about the relationship between social prescribing and social capital. As authors mentioned, “social prescribing refers to a means by which healthcare professionals seek to address the non-medical causes of ill health with non-medical interventions”. However, the social network that social capital emphasizes does not necessarily correspond to medical services. Thus, the connotations of social prescribing and social capital need to be elaborated more clearly.

Response:

Thank you for the constructive suggestions. We agree with the suggestions and have added evidence showing the relationship between social capital and health conditions in the introduction (lines 44 to 73).

  1. In the statistical analysis, the authors did not pay enough attention to the heterogeneity of network members. Is there significant difference between male and female groups, or various age groups? If the answer is yes, why gender/age difference occurs? The robustness test of the research results (e.g., harmonic closeness centrality and eccentricity decreased, and modularity increased, the network density in rural communities decreased gradually, and the diameter became smaller) is suggested to be added in order to improve the persuasiveness of the statistical evidence.

Response:

Thank you for the constructive suggestions. We agree with the suggestions and have added the analysis of gender, age, provision of Osekkai, and living places of the participants. In addition, we have added the description of the discussion regarding these analyses in the discussion part (Line 193 to 196 and 217 to 227).

In summary, this paper is well organized and provides some original findings. However, there are questions regarding the literature review, clarification of core concepts and analytical strategy.

Response:

Thank you for the constructive suggestions. We agree with the suggestions and have revised the suggested parts by adding a literature review and gender and age differences analysis.

Reviewer 2 Report

This work studies the possible relationship between social prescribing and social connections. The case study is Osekkai's conferences, which are used as a social prescribing tool. The analysis takes place through Social Network Analysis (SNA). The work is well written and very interesting.

However, there are some points where the work can be improved. For example, to fully understand what Osekkai's lectures were, I had to read about Ohta and Yata (2020), in which a concrete example of how these events work is expressed. I would also add a little more literature to support the motivations that led the authors to this study.

I did not know the methodology adopted. However, the authors offered a fairly comprehensive overview of the approach used. It was really helpful for me that each step was explained in detail.

The results demonstrate that there are no direct connections between Osekkai's conference and the increase in social connections. However, the authors explain well that these events cause indirect connections in the networks that increase the social capital of the community. Furthermore, it would be interesting to replicate the results in other areas of the country, or even in other geographical and cultural realities.

I consider the authors' work very interesting, I wish them success in their studies.

Author Response

Responses to the reviewers’ comments

Thank you very much for reviewing our manuscript and providing suggestions for its improvement. We have provided point-by-point responses to the reviewers’ comments; our revisions are indicated in red font here and in the document. We hope that the revised manuscript meets the journal’s requirements and can now be considered for publication.

This work studies the possible relationship between social prescribing and social connections. The case study is Osekkai's conferences, which are used as a social prescribing tool. The analysis takes place through Social Network Analysis (SNA). The work is well written and very interesting.

However, there are some points where the work can be improved. For example, to fully understand what Osekkai's lectures were, I had to read about Ohta and Yata (2020), in which a concrete example of how these events work is expressed. I would also add a little more literature to support the motivations that led the authors to this study.

Response:

Thank you for the constructive suggestions. We agree with the suggestions and revised the explanation of Osekkai conferences in the method section (Line 88 to 113).

I did not know the methodology adopted. However, the authors offered a fairly comprehensive overview of the approach used. It was really helpful for me that each step was explained in detail.

Response:

Thank you for the constructive suggestions. We agree with the suggestions and have revised the analysis of the background information (Line 193 to 196 and 217 to 227).

The results demonstrate that there are no direct connections between Osekkai's conference and the increase in social connections. However, the authors explain well that these events cause indirect connections in the networks that increase the social capital of the community. Furthermore, it would be interesting to replicate the results in other areas of the country, or even in other geographical and cultural realities.

Response:

Thank you for the constructive suggestions. We agree with the suggestions and have added the suggested issue to the limitation part (Line 289 to 300).

I consider the authors' work very interesting, I wish them success in their studies.